# The Eastern Caribbean Health Outcomes Research Network (ECHORN) Cohort Study: Design, Methods, and Baseline Characteristics

**DOI:** 10.3390/ijerph21010017

**Published:** 2023-12-21

**Authors:** Terri-Ann M. Thompson, Mayur M. Desai, Josefa L. Martinez-Brockman, Baylah Tessier-Sherman, Maxine Nunez, O. Peter Adams, Cruz María Nazario, Rohan G. Maharaj, Marcella Nunez-Smith

**Affiliations:** 1Ibis Reproductive Health, Cambridge, MA 02142, USA; 2Department of Chronic Disease Epidemiology, Yale School of Public Health, New Haven, CT 06520, USA; mayur.desai@yale.edu; 3General Internal Medicine, Yale School of Medicine, New Haven, CT 06520, USA; baylah.tessier@yale.edu (B.T.-S.); marcella.nunez-smith@yale.edu (M.N.-S.); 4School of Nursing, The University of the Virgin Islands, St. Thomas 00802, U.S. Virgin Islands; mnunez@uvi.edu; 5Faculty of Medical Sciences, The University of the West Indies, Cave Hill, Bridgetown P.O. Box 64, Barbados; peter.adams@cavehill.uwi.edu; 6Medical Sciences Campus, The University of Puerto Rico, San Juan 00921, Puerto Rico; cruz.nazario@upr.edu; 7Faculty of Medical Sciences, The University of the West Indies, St. Augustine, St. Augustine, Trinidad and Tobago; rohan.maharaj@sta.uwi.edu

**Keywords:** noncommunicable diseases, chronic disease, United States, Caribbean region, US territories, prevalence, health disparities, biobank, cohort study

## Abstract

Noncommunicable diseases (NCDs) account for a higher proportion of mortality and morbidity in the Caribbean and US territories—majority-minority communities—than in the United States or Canada. Strategies to address this disparity include enhancing data collection efforts among racial/ethnic communities. The ECHORN Cohort Study (ECS), a regional adult cohort study, estimates prevalence and assesses risk factors for NCDs in two United States territories and two Caribbean islands. Here, we describe the cohort study approach, sampling methods, data components, and demographic makeup for wave one participants. We enrolled ECS participants from each participating island using random and probability sampling frames. Data components include a clinical examination, laboratory tests, a brief clinical questionnaire, and a self-administered health survey. A subset of ECS participants provided a blood sample to biobank for future studies. Approximately 2961 participants were enrolled in wave one of the ECS. On average, participants are 57 years of age, and the majority self-identify as female. Data from the ECS allow for comparisons of NCD outcomes among racial/ethnic populations in the US territories and the US and evaluations of the impact of COVID-19 on NCD management and will help highlight opportunities for new research.

## 1. Introduction

Noncommunicable diseases (NCDs) are a leading cause of mortality and morbidity globally [1]. They account for 80.7% of all deaths in the Americas (35 countries, including the United States, Canada, and the Caribbean) [2]. The Caribbean region, which includes more than 25 countries and territories, has the highest NCD burden in the Americas [3]. Common NCDs that contribute to the high burden in the region include cardiovascular diseases, cancer, diabetes, and hypertension [3]. Study findings show a higher prevalence of diseases, such as diabetes and hypertension, in the Caribbean region compared to the United States (US) and Canada [4,5,6] and disparities in the provision of quality care in US territories compared to the US [7].

Many residents and citizens of the United States identify ancestrally and culturally with the US territories and other islands in the Caribbean. According to the 2021 American Community Survey, approximately 13 million US residents reported Caribbean and US territory ancestry [8]. Additionally, historically vulnerable patient populations in the United States share many characteristics with communities in the Caribbean [9,10,11]. Given the shifting demographics in the United States, attention to NCDs in ethnic minority populations can help advance the US agenda to reduce health disparities [12].

Addressing NCDs is a global priority. The Caribbean community’s concerns about the growing problem prompted the first United Nations General Assembly high-level meeting on NCDs [13] in 2011. At this meeting, leaders set goals for new global targets, an action plan for addressing NCDs, and obtaining more regional data to inform efforts to accurately assess the burden of NCDs and develop effective interventions [14]. Meanwhile, US engagement in global NCDs includes agencies such as the Centers for Disease Control and Prevention (CDC) establishing a division dedicated to global NCDs [15,16] and several National Institutes of Health as well as the Millennium Challenge Corporation (a US government corporation) supporting NCD surveillance, research, health projects that address NCDs, training programs, and strengthening country capacity [17,18,19]. Finally, the World Health Organization’s 2013–2020 Global Action Plan for the prevention and control of NCDs set an objective to promote research in low- and middle-income countries [20].

The National Institute on Minority Health and Health Disparities funds the Eastern Caribbean Health Outcomes Research Network (ECHORN), a collaboration among multiple academic institutions and community partners, to increase diversity in NCD research studies as well as bolster capacity for NCD data collection within the Caribbean region. ECHORN is a consortium of four academic institutions: the University of Puerto Rico Medical Sciences Campus in San Juan, the University of the Virgin Islands in St. Thomas and St. Croix, and the University of the West Indies Cave Hill and St. Augustine locations, in Bridgetown Barbados and Port of Spain Trinidad, respectively. The Coordinating Center is located at the Yale University School of Medicine.

ECHORN has a prospective longitudinal cohort study that aims to assess NCDs in four eastern Caribbean islands. The ECHORN Cohort Study (ECS) focuses on risk and protective factors across a wide range of chronic diseases and conditions and compliments cohort efforts on chronic disease in the region, across US territories, and in the United States [21,22,23,24,25]. It is particularly unique because of its broad NCD focus and multi-island partnerships. The data collected from the ECS have broad applicability because these data assess NCDs in a diverse and multi-ethnic community and include populations that account for a significant percentage of the immigrant population in the United States [26]. The ECS collects data, beginning in 2013, from majority racial/ethnic populations, offering an additional comparison group for racial/ethnic minority populations living in the United States and an ability to generalize research findings to diverse populations.

Here, we present a description of the sampling methods, data components, and baseline sample for wave one of the ECS. Data from the ECS may be used to characterize NCD risk and prevalence for cancer, diabetes, and cardiovascular disease for two US territories and two eastern Caribbean islands. Additionally, analyses of the ECS data may contribute to key research agendas, such as patient adherence and self-management, promoting health-related quality of life, and greater information on NCD practice, care, and outcomes [27].

## 2. The ECHORN Cohort Study (ECS)

The ECS, a prospective cohort, comprises Caribbean residents aged 40 years and older from the US territories of Puerto Rico (PR) and the US Virgin Islands (USVI), and independent Caribbean nations, including Barbados (BB) and Trinidad and Tobago (TT). PR is the largest island in the study with a population of 3.2 million, followed by TT (1.5 million), BB (281,835) and the USVI (99,076) [28,29,30,31]. The islands vary in demographics. The official language in BB, TT, and the USVI is English, while PR uses both English and Spanish. In TT and BB, healthcare is free and covered by the government and taxpayers. In contrast, the USVI and PR do not provide universal coverage and have a mix of public and privately financed health insurance plans. Additionally, the gross domestic product per capita in 2019 was $16,000 USD for TT, $18,000 USD for BB, $33,000 USD for PR, and $39,000 USD for the USVI.

The ECS provides (1) an estimated prevalence of diabetes, cancer, and heart disease for the cohort; (2) assesses the effect of known and potential risk factors associated with the development of diabetes, cancer, and heart disease; and (3) explores genetic determinants of NCDs.

## 3. A Collaborative Approach

Establishing a multi-site longitudinal cohort study requires a collaborative approach to protocol development, implementation, data analysis, and dissemination. At the start, we put together research teams in each island, including a principal investigator, faculty fellow, project manager, research assistant, and clinical research nurse. The US-based coordinating center houses staff who provide overall administrative coordination as well as technical expertise. Our research teams and coordinating center work together to acquire ethics approval for the ECS and to design research procedures and instruments. This collaborative approach ensures that the research measures are appropriately adapted to each location and increases the fidelity of research implementation.

We employ a community-engaged research approach throughout the study design, implementation, and translation phases. We work with local researchers and community members to (a) identify and collaborate with local statisticians; (b) source local sampling surveys to create sampling strategies specific to each research site; (c) establish community advisory boards; (d) observe local customs related to research study implementation (such as one site not incorporating participant remuneration); (e) establish research centers in communities that are easily accessible by public transportation; and (f) assess the acceptability of the ECS through the conduct of a pilot study with community members prior to launch.

## 4. Methods

### 4.1. Sampling Strategy

For wave one data collection, we recruit a sample of residents in each of the four island sites. We create sampling frames using national surveys in each island as a basis: The Centers for Disease Control (CDC) Behavioral Risk Factor Surveillance System in the US Virgin Islands; the Barbados Health of the Nation Study and Barbados Labour Surveys; the Central Statistical Office Continuous Household Surveys in Trinidad; and the US Census Track data in Puerto Rico. The ECS sample size is based on the population size of each island, with the intention that larger island sites will contribute a greater proportion of participants to the sample.

Puerto Rico, Barbados, and Trinidad use a stratified multi-stage probability sample design, where residents come from households within randomly selected enumeration districts. In Barbados, we sample residents from 25 randomly selected enumeration districts across the entire island. In Puerto Rico and Trinidad, we sample residents from households within 96 and 70 randomly selected enumeration districts, respectively, in select communities/municipalities in the island. These select communities/municipalities have distributions of age, race/ethnicity, sex, and education level similar to the island’s general population.

For the US Virgin Islands, we use a simple random sampling technique, such that each resident has an equal probability of being selected for the study. In the US Virgin Islands, we invite residents from St. Thomas, St. Croix, St. John, and Water Island to participate in the study by phone using a random digit dialing process.

For all island sites that use a stratified multi-stage probability sampling frame, a single participant is selected from each household. In Barbados, Trinidad, and Puerto Rico, we recruit residents through household visits during the day and evening and on weekday and weekends. In the US Virgin Islands, we recruit residents by phone. We make three attempts to reach an adult at the residence. If there is no response, the number/household is removed from the recruitment list. In the event of multiple eligible residents, we invite the adult with the most recent birthdate. If they decline participation, then we invite the next eligible resident from the household to participate.

The sampling strategy and recruitment protocols for each island site is provided in Appendix B.

### 4.2. Eligibility Criteria

We require ECS participants to be 40 years of age or older, live in one of the four islands, speak English or Spanish, and have a reliable contact/residential address and a permanent or semi-permanent resident for at least 10 years with no plans for permanent relocation within five years. Importantly, we do not consider participants’ health status at the time of enrollment when determining eligibility. We verify a resident’s ability to provide informed consent and complete the baseline assessment, by asking them to complete a cognitive test, the mini-cog^4^. We require a score of 1, 2, or 3 with a normal clock drawing test, indicating no cognitive impairment, for enrollment. Once a resident agrees to join the study, we schedule an appointment to visit a community assessment center. Once the baseline assessment is complete, participants in the USVI, TT, and PR are given USD$25 in a gift voucher or cash. Following local guidance, we do not provide incentives to participants from the Barbados site. Wave one participants join the ECS starting in June 2013 and ending in June 2018. Follow-up begins 4–5 years after each participant’s baseline assessment. As participants enter the follow-up window, they are identified and contacted (Figure 1).

#### Data Elements

The ECS harmonizes with other US cohort studies, such as the Jackson Heart Study [21] and Hispanic Community Health Study/Study of Latinos [23], by integrating similar clinical, behavioral, and demographic measures into its design. Data components include a clinical examination, laboratory tests, a brief clinical questionnaire, and a self-administered health survey. The ECS biobank stores genetic information for a subset of participants. The biobank aims to accelerate the identification of new markers of disease and environmental factors associated with NCDs. We invite participants from three of four island sites where biobanking is approved by local institutional review boards to provide a biological specimen for biobanking.

### 4.3. Clinical Exam and Laboratory Tests

We ask wave one ECS participants to complete a clinical examination, which begins with a family history, including a report of NCDs, other medical conditions, and deaths in the extended family. They are also asked to report all medications that they are taking at enrollment and to describe medication adherence. Anthropometric measurements are captured next by trained clinical research nurses. Measures include weight; height; skin fold thickness; waist, hip, and neck circumference; grip strength; resting clinic blood pressure; and an ankle brachial index. All wave one participants provide a blood sample at the beginning of their clinical assessment for immediate testing. Blood tests for lipids (fasting or non-fasting), a complete blood count, hemoglobin A1C, and a comprehensive metabolic panel are performed. See Table 1 for details. Given the geographical distance between island sites, no central laboratory is used to process blood samples. We use point of care machines for blood sample processing at 3 of our 4 island sites. In Puerto Rico, we use a clinical laboratory at the Hispanic Alliance for Clinical & Translational Research in Puerto Rico (formerly the Puerto Rico Clinical and Translational Research Center). Participants can share the results of laboratory tests with their primary care providers.

### 4.4. Self-Reported Health Survey

We also ask wave one ECS participants to complete a health survey in English or Spanish using a keyboard or touch screen. We use the Questionnaire Development System (QDS) software (Nova Research, version 2.6.1.2) to collect baseline survey data. Audio computer assisted self-interviewing (ACASI) is made available to those participants who want the survey read aloud to them. For those participants who are not comfortable with the computer-assisted interview, we allow a member of the research team to read the survey aloud to them. In addition to demographic variables, such as age, sex, marital status, employment, and sexual orientation, we collect data on several factors potentially associated with NCDs. Alcohol use, smoking, physical activity, sleep habits, reproductive history, access to care, chronic disease history, depressive symptoms, and migration history are some of the factors collected. A modified version of the National Health and Nutrition Examination Survey (NHANES) Dietary Screening Questionnaire (DSQ) [32] is used to assess nutritional intake. Finally, we ask all participants to provide information on the most influential members of their health network as it relates to health information, advice, and health decisions. Participants vary in terms of survey completion, with most taking between 60 and 180 min. Table 1 outlines the baseline data by collection method.

## 5. Data Quality Assurance Procedures

To ensure consistency in data collection across our island sites, the medical director at the coordinating center trains all clinical research nurses on the clinical procedures at an in-person meeting (June 2012) and offers refreshers through online videos and training sessions. Further, a manual of operations, which offers visual depictions, detailed descriptions of each clinical measure, and instructions for each procedure along with special considerations and potential modifiers to facilitate proper measurement, is shared with the clinical research staff. All other research staff receive training from the coordinating center on the research protocols (between June 2012 and May 2013). Additionally, each island site pilots the ECS baseline protocol, from intake (inclusive of laboratory testing) to exit with 3–4 community members. This ensures that research staff are confident in the process at launch, but also provides an opportunity to resolve issues with the research protocol. In Puerto Rico, only certified clinical research nurses employed by the Alliance collect the clinical observations (see Table 1) and complete the blood draw. At all other research sites, clinical research nurses are part of the research team, performing the clinical examinations and phlebotomy. Laboratory data are entered into a REDCap database. Data from all sources are doubly entered, cleaned, and merged into SAS (SAS Institute, Carey, NC, USA). Data quality assurance, storage, cleaning, and analysis are conducted at the coordinating center. Greater details on the clinical data in the ECS can be found in Appendix A.

Requests for data and data analysis are managed by the Data Access and Scientific Review (DASR) committee. Access to the manual of procedures is contingent on approval from the DASR committee. The DASR committee is made up of the five principal investigators, the ECHORN Coordinating Center Director (Chair; non-voting member), and ad-hoc members with expertise in epidemiologic study design and biostatistics (non-voting members).

## 6. Results

### 6.1. Description of the ECS Wave One Sample

In total, wave one of the ECS has 2961 participants, reflecting an overall participation rate of 70%. Participation at all island sites is above 50% (Table 2).

The majority of wave one participants identify as female and are on average 57 years old (range 40–91). Close to 62% of the sample rate their general health as ‘good’ or ‘very good’. Those who did not enroll in the cohort were slightly younger (56 vs. 57 years, *p* < 0.05) and more likely to be male (39% vs. 35%, *p* < 0.05). See Table 3 for demographic and health characteristics of ECS wave one participants.

### 6.2. The Biobank

Three of the four island sites are a part of the biobank: PR, BB, and the USVI. We divide our blood samples into fifteen 0.5 mL aliquots, store them temporarily at −80 °C on-site, and then ship them to the coordinating center for permanent storage. All aliquots are stored at the coordinating center because of concerns power and back-up power to the −80 °C freezers in the island sites. Biobank samples are stored for a short time in-island and then sent, using dry ice, to the US-based coordinating center for long-term storage. We divide the biobank samples into multiple aliquots to ensure many investigators can use the samples to examine environmental chemicals and nutritional biomarkers and draw causal links to disease.

A total of 635 participants are included in the biobank: 10% of participants in BB (n = 101), 27% in the USVI (n = 95), and 57% in PR (n = 439). Nearly two-thirds (65%) of the biobank participants are female, and approximately 64% of biobank samples are from participants ages 50–69 years. About 15% of the samples are from participants 70 years and older. The biobank complements efforts already underway in the United States, such as the National Biomonitoring Project at the CDC, to better understand factors associated with NCD development. The number of samples by type and island site are shown in Table 4.

## 7. Discussion

The Caribbean and US territories in the Caribbean Sea as a region is home to diverse multiethnic communities that bear many similarities with populations in the US. However, a paucity of health outcomes research on Caribbean and Caribbean diasporic populations limits our understanding of the many disparities that affect these populations. The ECS, a research initiative from a NIMHD-funded consortium, bolsters efforts to conduct NCD research with racial/ethnic minorities and strengthens workforce capacity to monitor NCD prevalence and risk. The ECS is particularly relevant given increased attention to NCD prevention and management following the COVID-19 pandemic [2]. The ECS will offer data on the impact of the pandemic on NCDs in this population, help identify points for intervention, and highlight opportunities for new research.

The ECS complements ongoing cohort research in the United States by incorporating similar clinical, behavioral, and demographic measures. Further, it offers the opportunity to compare health outcomes across multiple groups within the cohort and across cohorts. Using data from the ECS, researchers will be able to compare health outcomes for the US territories versus non-US Caribbean sites, compare the ECS to extant datasets focused on racial/ethnic populations in the US, and compare the ECS with emerging datasets and projects.

In addition to providing information on the minimum NCD indicators (nutrition, alcohol and tobacco use, physical activity, blood pressure, weight, cholesterol, and healthcare seeking), the ECS collects information on novel risk/protective factors, such as sleep and use of alternative medicine, and can assess the role of health networks on health behaviors. Further, the rich diversity of the ECS sample allows researchers to stratify risks by several key social characteristics, such as income, gender, and racial/ethnic self-identification. Finally, the ECS biobank facilitates the identification of new biomarkers for chronic diseases.

Another strength of the ECS lies in its application of a regional approach and its use of a community-engaged approach in the development and implementation of the study. A common challenge to NCD research efforts in the Caribbean region is that countries with limited resources have difficulty achieving economies of scale. Regional coordination allows for the collection of data on multiple NCDs at multiple sites. A second advantage of the regional approach is the ability to assess and highlight similarities and differences in NCD risk based on contextual and clinical factors across the eastern Caribbean. Finally, the regional approach facilitates health information exchange and the integration of interventions across multiple countries.

Community-engaged methods center the community’s assets and strengths and ensure that programs, policies, and studies that emerge are sustainable, effective, and relevant. We used site-specific statisticians and research teams as well as community advisory boards to promote the use of local knowledge and expertise for our sampling strategies, study instruments, and study implementation. Furthermore, the use of localized research teams to collect data and a standardized research protocol ensures the longevity of the study by supporting ancillary studies on NCDs in the same islands, expansion to other islands in the region, and research initiatives that respond to changes in the pattern of the NCD epidemic. As an illustration, in response to reports of increasing numbers of children being diagnosed with chronic diseases, such as diabetes and hypertension, ECHORN is now recruiting the children and grandchildren (ages 5–17 years) of existing ECS participants, establishing the first multi-island intergenerational cohort study in the region.

There are notable limitations to the ECS. Importantly, the sample size of the ECS does not allow us to generate prevalence estimates for each island in the study. Further, some clinical data elements may be missing for a portion of participants. Budgetary limitations led some island sites to shift from using reference laboratories to point of care machines. During that shift, blood sample collection was halted at those sites. Biobank samples are only available for three of the four island sites. Trinidad did not receive approval from the local institutional review board to contribute biospecimens to the biobank. Additionally, St. Croix did not have a −80 °C freezer on site to store samples, so biospecimen collection in the USVI is limited to St. Thomas. Finally, the effectiveness of recruitment efforts varied by island site as shown in Table 2.

## 8. Conclusions

The ECS, a regional cohort study, evaluates the prevalence of and risk factors for heart disease, diabetes, and cancer among adults residing in two US territories and two Caribbean islands. Clinical, behavioral, and demographic data are available for approximately 2961 participants. Further, the ECS stores biological specimens for 635 participants, which may aid in the identification and management of noncommunicable diseases. Data from the ECS are an important contribution to the United States’ efforts to reduce health disparities because it offers insights into biological, clinical, and behavioral factors associated with NCD health outcomes among racial/ethnic minorities. Considering the growing proportions of the US population reporting Caribbean lineage, analyses of ECS data will shed light on similarities between US residents and those in the US territories and signal contextual differences that may place residents in the US at greater risk for NCDs. The ECS is the flagship cohort study of ECHORN. ECHORN welcomes requests for ancillary studies, data analyses, and scientific collaborations using the ECS data and research infrastructure. The second wave of data collection (first wave of follow-up) is nearly complete, and we are actively planning for wave three.

## Figures and Tables

**Figure 1 ijerph-21-00017-f001:**
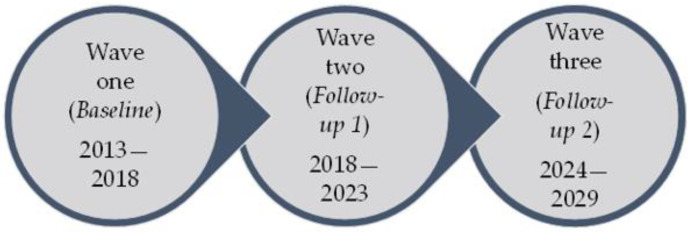
ECS Data Collection Timeline.

**Table 1 ijerph-21-00017-t001:** Components of ECS wave one data collection.

Components	Clinical Interview	Clinical Exam	Questionnaire
**Clinical Observations**			
Medical History	X		
Social Network	X		
Medication Use & Adherence	X		
Height and weight		X	
Waist, hip, and neck circumference		X	
Grip strength		X	
Resting blood pressure		X	
Ankle brachial index		X	
*Laboratory tests*			
Lipid panel		X	
Complete blood count		X	
Hemoglobin A1C		X	
Metabolic panel		X	
**Survey Elements**			
Demographics (e.g., age, sex, gender identity, marital status, parity, etc.)			X
Global Health (general health assessment)			X
*Physical and Mental Health*			
History of chronic diseases			X
Health care access and utilization			X
Reproductive health			X
Depression			X
*Health behaviors*			
Substance use			X
Tobacco use			X
Physical activity			X
Nutritional assessment			X
Sleep			X
*Social Health*			
Incarceration			X
Social support			X
Neighborhood factors			X

**Table 2 ijerph-21-00017-t002:** ECS wave one participants enrolled by island site and recruitment method.

Recruitment Method	Site	Encountered	Eligible	Enrolled	Participation Rate
Household	BB	2317	1415	1008	71%
Household	PR	1192	988	771	78%
Household	TT	1326	1235	829	67%
Random Digit Dialing	USVI	1022	618	353	57%
	Total	5857	4256	2961	70%

**Table 3 ijerph-21-00017-t003:** ECS wave one participant characteristics (N = 2961).

Characteristic	N (Available Data)	Mean (SD)
*Age* (years)	2961	57.3 (10.3)
*BMI* (Kg/m^2^)	2923	29.2 (6.1)
*Total cholesterol* (mmol/L)	2372	193 (39.3)
*HDL* (mmol/L)	2345	52.6 (14.6)
		**N (%)**
*Sex*, Female	2961	1935 (65.4)
*Age* (years)	2961	
40–49		742 (25.1)
50–59		1046 (35.3)
60–69		775 (26.2)
70+		398 (13.4)
*Marital status*	2901	
Single		1181 (40.7)
Married		1196 (41.2)
Divorced or separated		304 (10.5)
Widowed		220 (7.6)
*Sexual orientation*	2792	
Heterosexual (‘straight’)		2488 (89.1)
Gay or lesbian		25 (0.9)
Bi-sexual		17 (0.6)
Not sure or questioning		262 (9.4)
Regular source of care (*Is there one place you usually go when you need routine or non-emergency care?*)	2934	
Yes		2093 (71.3)
I do not seek routine care anywhere		405 (13.8)
I seek routine care at more than one place		436 (14.9)
General Health (*In general, would you say your health is:*)	2956	
Poor		84 (2.8)
Fair		800 (27.1)
Good		1267 (42.9)
Very Good		545 (18.4)
Excellent		260 (8.8)
*Comorbidities*		
Obese (BMI 30+)	2923	1136 (38.9)
Hypertension ^a^	2953	1459 (49.4)
Cardiovascular Disease ^b^	2956	410 (13.9)
Diabetes ^c^	2503	562 (22.5)
Cancer	2953	121 (4.1)
Current Smoker	2843	230 (8.1)
*At least one of the comorbidities above*	2829	2213 (78.2)
*PAD* (ABI ≤ 0.9)	2772	121 (4.4)

^a^ Hypertension is self-reported. Baseline rate also includes borderline or pre-hypertension. ^b^ Cardiovascular disease is self-reported and includes coronary heart disease, angina, abnormal heart rhythm, heart attack, stroke, or congestive heart failure. ^c^ Diabetes is self-reported.

**Table 4 ijerph-21-00017-t004:** The ECS biorepository by type of sample and island site.

	All Sites		BB		PR		USVI	
Biobank participants	635		101		439		95	
Sample type	n	%	n	%	n	%	n	%
Blood clot	633	5.93	104	5.99	435	5.87	94	6.13
Serum	3098	29.03	472	27.20	2201	29.72	425	27.72
Whole blood	6942	65.04	1159	66.80	4769	64.40	1014	66.14

## Data Availability

The data presented in this paper are available on request from the corresponding author. Requests for ECS data and/or analyses of ECS data may be requested through the Data Access and Scientific Review (DASR) committee (https://www.echorn.org/request-echorn-data).

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
