# Peer review of "The Eastern Caribbean Health Outcomes Research Network (ECHORN) Cohort Study: Design, Methods, and Baseline Characteristics"

_ijerph, 2023, doi:10.3390/ijerph21010017_

Round 1

Reviewer 1 Report

Comments and Suggestions for Authors

This is a very well-written and thorough description of the ECHORN Cohort Study (ECS) that will be a terrific reference for manuscripts based on the study.  This cohort addresses important study populations, has a very well thought out study design, utilized community engagement approaches to help establish the study and ensure its success, has the ability to compare findings to those in other populations due to similarity of methods and instruments used, and has other positive features.  I applaud the study team for constructing such a well-designed study.  My suggestions for improvement of the manuscript are to:

·       Add a sentence or two about the follow-up plans for the cohort.

·       Add some basic information about the characteristics of the non-responders

·       Include information about which study site does not have biospecimens banked for later studies

·       Include information about the rationale for the size of the cohort in the sites (what drove the decisions?) and address any strengths or limitations of those numbers in the discussion.  

The Appendix is written in the future tense, whereas the cohort ascertainment and baseline data has already collected. For example, page 10 line says “ECHORN Barbados aims to recruit 1,100 participants aged 40 and older”. Or page 10 lines 375-380 which outlines several possible sampling scenarios.  It reads like it is the sampling statisticians’ proposal for the study sample.  Some editing is needed to reflect what happened, not just want was planned. Otherwise, the detail provided in the Appendix is great and appropriate.

Is the phrase “the ECS” missing in page 10 line 371 before “but requires greater numbers of participants”?

On page 6 line 230 the word biostatistics is misspelled

Author Response

Recommendations

Responses

This is a very well-written and thorough description of the ECHORN Cohort Study (ECS) that will be a terrific reference for manuscripts based on the study.  This cohort addresses important study populations, has a very well thought out study design, utilized community engagement approaches to help establish the study and ensure its success, has the ability to compare findings to those in other populations due to similarity of methods and instruments used, and has other positive features.  I applaud the study team for constructing such a well-designed study.  My suggestions for improvement of the manuscript are to:

Thank you for your kind comments, we are hopeful that this collaborative NCD study will be an example to other researchers.

Add a sentence or two about the follow-up plans for the cohort.

Thank you for this recommendation. We have included the following sentence in conclusion: “The second wave of data collection (first wave of follow-up) is nearly complete, and we are actively planning for wave 3.”

Add some basic information about the characteristics of the non-responders

Thank you for catching this oversight. We have added the following text to the manuscript under the header ‘description of the ECS wave one sample’: Those who did not enroll in the cohort were slightly younger (56 vs 57, p<0.05) and more male (39% vs. 35%, p<0.05).”

Include information about which study site does not have biospecimens banked for later studies

We appreciate this recommendation. Under header “the biobank”, we did name the three country sites that have biospecimens banked for later studies.  However, we have included a sentence in the limitations section that reads:  “Biobank samples are only available for three of the four country sites. Trinidad did not receive approval from local institutional review board to contribute biospecimens to the biobank. Additionally, St. Croix did not have a -80° Celsius freezer on site to store samples, so biospecimen collection in the U.S. Virgin Islands is limited to St. Thomas.

Include information about the rationale for the size of the cohort in the sites (what drove the decisions?) and address any strengths or limitations of those numbers in the discussion  

In the methods section, under the header “ sampling strategy”. We state: “The ECS sample size is based on the population size of each country, with the intention that larger country sites will contribute a greater proportion  of participants to the sample.

In the limitations and strengths section, we have added the following sentence:

“The effectiveness of recruitment efforts varied by country-site as shown in Table 2.”

The Appendix is written in the future tense, whereas the cohort ascertainment and baseline data has already collected. For example, page 10 line says “ECHORN Barbados aims to recruit 1,100 participants aged 40 and older”. Or page 10 lines 375-380 which outlines several possible sampling scenarios.  It reads like it is the sampling statisticians’ proposal for the study sample.  Some editing is needed to reflect what happened, not just want was planned. Otherwise, the detail provided in the Appendix is great and appropriate

Thank you for this observation. What is included in the appendix is what was done; however, we see how the tense-use could make that confusing.  We have reviewed all appendices and updated the tense. Additionally, we have updated the sample size (i.e. what was collected vs. what was intended). 

Is the phrase “the ECS” missing in page 10 line 371 before “but requires greater numbers of participants”?

We have located this phrase within appendix B under the sample design description for the Barbados country-site.  We have updated the phrase to read:

“Previous surveys (notably the rolling Barbados Labour Force Survey and the ongoing 2012 Barbados Behavioral Risk Factor Survey - The Health of the Nation) have each selected 45 EDs for survey recruitment but require a greater number of participants than the ECS.”

On page 6 line 230 the word biostatistics is misspelled

Thank for you catching this spelling error, we have since corrected it.

Reviewer 2 Report

Comments and Suggestions for Authors

The authors present the description of a new cohort study (ECHORN), with a unique trait of having enrolled participants belonging to racial/ethnic groups which are historically under-represented in research. This article describes the design and baseline characteristics of this cohort. Data from this cohort have been analyzed and published in previous studies; however, this appears to be the first report dedicated to describing the cohort design and construction process. This article can potentially serve as a useful resource for prospective researchers interested in analyzing data from this cohort. The report is comprehensive and succinct, however, the presentation appears a bit hard to follow. Suggestions for improvement:

1) The study timeline for baseline enrollment and follow-up waves is unclear. The participant enrollment process was prolonged between 2013-2018 (5 years): please could you provide a figure for the timeline showing how/why it took 5 years to complete enrollment? Most other studies complete enrollment in a few years. The prolonged enrollment is concerning particularly because the authors said the next wave (follow-up) was done 2 years after wave 1... does this mean 2 years after the participant's baseline visit or after completion of the first/baseline wave? if wave 1 is 5 years long and wave 2 was done in 2020, then participants will have variable follow-up, rendering the validity of longitudinal analyses questionable. For an example of a figure, please see following websites:

https://aric.cscc.unc.edu/aric9/about/project_overview

https://www.mesa-nhlbi.org/MESA_508TextOnly.htm

2) Please could you dedicate a separate paragraph (with its own subheading) for how readers can access the cohort data? Is the dataset available in public repositories such as NHLBI's BioLINCCC?

3) Table 4 data is very unclear/confusing. If there are only 2961 participants in the cohort, why are the total n for different sample types variable (635 - 6942)? Please mention n for number of participants as separate columns in that table.

4) Please provide more details on how anthropometric and clinical data elements mentioned in table 1 were measured (definitions, instruments, 'n' ie number of participants for each element, a link to the manual of operations), were assessors centrally trained or were there important differences in measurement techniques across countries?

5) Please discuss limitations of the cohort in a detailed/comprehensive manner. Limitations for such a large cohort study are often best known only to the PIs and study team. Sharing this information with readers can help them plan their use/secondary analyses of these data more rigorously.

6) In table 3, please provide more information about baseline characteristics. Information about race of participants is unclear- was this data collected/confirmed? was the race of all participants confirmed to be the same? Please provide comorbidity burden and descriptive statistics for lab values and other clinical measures at baseline.

Thank you for your contributions.

Comments on the Quality of English Language

n/a

Author Response

The authors present the description of a new cohort study (ECHORN), with a unique trait of having enrolled participants belonging to racial/ethnic groups which are historically under-represented in research. This article describes the design and baseline characteristics of this cohort. Data from this cohort have been analyzed and published in previous studies; however, this appears to be the first report dedicated to describing the cohort design and construction process. This article can potentially serve as a useful resource for prospective researchers interested in analyzing data from this cohort. The report is comprehensive and succinct, however, the presentation appears a bit hard to follow. Suggestions for improvement:

Thank you for your recommendations to refine our manuscript. We are hopeful, as you indicated, that this manuscript will be a helpful resource.

1) The study timeline for baseline enrollment and follow-up waves is unclear. The participant enrollment process was prolonged between 2013-2018 (5 years): please could you provide a figure for the timeline showing how/why it took 5 years to complete enrollment? Most other studies complete enrollment in a few years. The prolonged enrollment is concerning particularly because the authors said the next wave (follow-up) was done 2 years after wave 1... does this mean 2 years after the participant's baseline visit or after completion of the first/baseline wave? if wave 1 is 5 years long and wave 2 was done in 2020, then participants will have variable follow-up, rendering the validity of longitudinal analyses questionable. For an example of a figure, please see following websites:

https://aric.cscc.unc.edu/aric9/about/project_overview

https://www.mesa-nhlbi.org/MESA_508TextOnly.htm

Baseline enrollment took 5 years (2013-2018). This is comparable to other cohort studies: Jackson Heart Study baseline enrollment took 4 years (2000-2004) and Framingham Heart Study baseline enrollment also took 4 years (1948-1952).

Wave 2 (the first follow-up assessment) began in 2018. Follow-up was designed to take place 4-5 years after each participant’s baseline assessment. Follow-up occurs on a rolling basis, such that as participants enter the follow-up window, they are identified and contacted. Cohort studies, by design, have varying amounts of person-time for each participant and this will be taken into account in any analyses.

In terms of the manuscript, we have updated appendix B, removing the incorrect sentence in the Barbados description about a biennial follow-up.

Additionally, we have included two sentences describing follow-up at the end of the section titled “eligibility criteria” and added a figure as you recommended.

The sentences read: “ Follow-up begins 4-5 years after each participant’s baseline assessment. As participants enter the follow-up window, they are identified and contacted.”

2) Please could you dedicate a separate paragraph (with its own subheading) for how readers can access the cohort data? Is the dataset available in public repositories such as NHLBI's BioLINCCC?

We agree that readers should know how to access the dataset. The journal asks that we include this information in a data availability statement, which is included at the end of the article. In our data availability statement, we state:

“The data presented in this paper are available on request from the corresponding author. Requests for ECS data and/or analyses of ECS data may be requested through the Data Access and Scientific Review (DASR) committee (https://www.echorn.org/request-echorn-data). “

Additionally, text related to data availability were included under section header: Data quality assurance procedures.  That text read: “Requests for data and data analysis are managed by the Data Access and Scientific Review (DASR) committee. The DASR is made up of the five principal investigators, the ECHORN Coordinating Center Director (Chair; non-voting member), and ad-hoc members with expertise in epidemiologic study design and biostatistics (non-voting members).”

To that text, we have added the line: “ Access to the manual of procedures if contingent on approval from DASR

3) Table 4 data is very unclear/confusing. If there are only 2961 participants in the cohort, why are the total n for different sample types variable (635 - 6942)? Please mention n for number of participants as separate columns in that table.

Thank you for calling this confusion to our attention. In the text, under the header “ the biobank” we indicate that blood samples are divided into 15 aliquots, as such, each individual contributes multiple samples. The table is intended to illustrate the abundance of samples available by type and country site.

While we did include the number of participants that contributed by country-site to the biobank within the text, we have now added that information to the table as well.

4) Please provide more details on how anthropometric and clinical data elements mentioned in table 1 were measured (definitions, instruments, 'n' i.e. number of participants for each element, a link to the manual of operations), were assessors centrally trained or were there important differences in measurement techniques across countries?

Thank you for these detailed suggestions.

We have included more information on the anthropometric and clinical data elements in the text as well as with a second appendix (now labeled Appendix A). Appendix A contains details on the n, definitions, and equipment used for the clinical and anthropometric data elements mentioned in table 1.

Under section header ‘Data Quality Assurance Procedures’ we included the following text  “To ensure consistency in data collection across our country sites, we train all research staff on the clinical and research protocols prior to launch (between June 2012 and May 2013)”. We have updated that text to now read: “ […] the medical director at the coordinating center trains all clinical research nurses on the clinical procedures at an in-person meeting in June 2012 and offers refreshers through online videos and training sessions. Further, a manual of operations, which offers  visual depictions, detailed descriptions of each clinical measure, and instructions for each procedure along with special considerations and potential modifiers to facilitate proper measurement, is shared with the clinical research staff .

All other research staff receive training from the coordinating center on the research protocols between June 2012 and May 2013.” 

We have also included a sentence that reads: “Greater details on the clinical data in the ECS can be found in Appendix A.”

5) Please discuss limitations of the cohort in a detailed/comprehensive manner. Limitations for such a large cohort study are often best known only to the PIs and study team. Sharing this information with readers can help them plan their use/secondary analyses of these data more rigorously.

We appreciate your recommendation to add more detail to the strengths and limitations section to support researchers interested in using the ECS data. Throughout the manuscript, we have highlighted limitations in our recruitment and implementation. Nonetheless, we have included a few additional sentences in the limitations section. Specifically:

Biobank samples are only available for three of the four country sites. Trinidad did not receive approval from the local institutional review board to contribute biospecimens to the biobank. Additionally, St. Croix did not have a -80° Celsius freezer on site to store samples, so biospecimen collection in USVI is limited to St. Thomas. Finally, the effectiveness of recruitment efforts varied by country site as shown in Table 2.”

6) In table 3, please provide more information about baseline characteristics. Information about race of participants is unclear- was this data collected/confirmed? was the race of all participants confirmed to be the same? Please provide comorbidity burden and descriptive statistics for lab values and other clinical measures at baseline.

We have updated table 3 to include additional baseline characteristics. It now includes some clinical measures at baseline, as well as a measure of comorbidity burden.

In concert with a recommendation from the Community Advisory Boards, the Principal Investigators for ECHORN have decided not to include race/ethnicity data in Table 1 or in analyses. The race/ethnicity variable is not commonly used in the Caribbean region, would not be used to represent social experiences that help explain NCD risk, and are not being used to address questions directly related to these concepts.  (see  Ross et al. Considerations for using race and ethnicity as quantitative variable in medical education research (https://www.ncbi.nlm.nih.gov/pmc/articles/PMC7550522/) and best practices for using race in public health research (https://publichealth.uic.edu/community-engagement/collaboratory-for-health-justice/best-practices-race-public-health-research/  )

Round 2

Reviewer 2 Report

Comments and Suggestions for Authors

Thank you for addressing comments from the previous round of review. I have no additional comments. Thank you for your contributions.

Comments on the Quality of English Language

N/A